# Assessment of In Vitro Bioaccessibility and In Vivo Oral Bioavailability as Complementary Tools to Better Understand the Effect of Cooking on Methylmercury, Arsenic, and Selenium in Tuna

**DOI:** 10.3390/toxics9020027

**Published:** 2021-02-03

**Authors:** Tania Charette, Danyel Bueno Dalto, Maikel Rosabal, J. Jacques Matte, Marc Amyot

**Affiliations:** 1Groupe de Recherche Interuniversitaire en Limnologie (GRIL), Département de Sciences Biologiques, Université de Montréal, Complexe des Sciences, C.P. 6128, Succ. Centre-Ville, Montréal, QC H3C 3J7, Canada; tania.charette@umontreal.ca; 2Sherbrooke Research and Development Centre, Agriculture and Agri-Food Canada, Sherbrooke, QC J1M 0C8, Canada; danyel.buenodalto@canada.ca (D.B.D.); jacques.matte2@canada.ca (J.J.M.); 3Groupe de Recherche Interuniversitaire en Limnologie et en Environnement Aquatique (GRIL), Département des Sciences Biologiques, Université du Québec à Montréal (UQAM), 141 Avenue du Président-Kennedy, Montréal, QC H2X 1Y4, Canada; rosabal.maikel@uqam.ca

**Keywords:** methylmercury, arsenic, selenium, oral bioavailability, bioaccessibility

## Abstract

Fish consumption is the main exposure pathway of the neurotoxicant methylmercury (MeHg) in humans. The risk associated with exposure to MeHg may be modified by its interactions with selenium (Se) and arsenic (As). In vitro bioaccessibility studies have demonstrated that cooking the fish muscle decreases MeHg solubility markedly and, as a consequence, its potential absorption by the consumer. However, this phenomenon has yet to be validated by in vivo models. Our study aimed to test whether MeHg bioaccessibility can be used as a surrogate to assess the effect of cooking on MeHg in vivo availability. We fed pigs raw and cooked tuna meals and collected blood samples from catheters in the portal vein and carotid artery at: 0, 30, 60, 90, 120, 180, 240, 300, 360, 420, 480 and 540 min post-meal. In contrast to in vitro models, pig oral bioavailability of MeHg was not affected by cooking, although the MeHg kinetics of absorption was faster for the cooked meal than for the raw meal. We conclude that bioaccessibility should not be readily used as a direct surrogate for in vivo studies and that, in contrast with the in vitro results, the cooking of fish muscle did not decrease the exposure of the consumer to MeHg.

## 1. Introduction

Fish provides a means of food security and offers an elevated nutritional value [1]. The consumption of fish is also, however, the primary pathway of human exposure to mercury (Hg) and its organic form, methylmercury (MeHg) [2]. Even chronic low MeHg exposure, for example at concentrations observed for average fish consumption in the Western Hemisphere, could be toxic for neurological, cardiac, and immune systems [3,4]. Currently in Canada, Hg is the only element on which fish consumption guidelines are based [5].

Methylmercury is the main Hg species found in fish muscle, representing over 80% of total Hg [6]. Along with high levels of MeHg, the muscle of fish such as tuna is also rich in selenium (Se) and arsenic (As) [7,8]. Antagonistic effects of Se on MeHg toxicity have been reported [9]. In fish muscle, Se can be found in inorganic or organic forms, such as selenate, selenomethionine, and selenoneine [7,10,11]. Further, there is a potential antagonist relationship between Se and As that could limit the protective effect of Se against MeHg [12]. However, the main As species known to interact with Se are the inorganic forms [12], this later representing a low proportion in fish muscle compared with that of organic species (iAs ranges from <2 to 30% of total As) [13,14,15,16]. In its organic form, As can be found as arsenobetaine (AsB) and arsenolipid [17]. Therefore, the study of MeHg behavior should include As as well since they are both interrelated by the metabolism of Se [18], and even more so since the identification of the Hg-Se-As complex by Korbas et al. [19].

In vitro digestion has emerged as a rapid, cheap, and highly replicable tool for assessing Hg exposure from fish consumption [20]. This technique consists of measuring the bioaccessibility of a given component, defined as the soluble fraction that is released from the food matrix by the action of digestive processes [21]. In vitro models are valuable but must be used with caution because they can fail to include several aspects of human digestive physiology. Pig-based studies have successfully validated the use of in vitro digestion as a surrogate of in vivo studies for iAs bioavailability from soils [22,23]. Because of its similar digestive system, the pig model is a relevant alternative that can be used as surrogate for humans [24]. In contrast, no study has yet confirmed the reliability of in vitro digestion of MeHg from fish muscle to evaluate its in vivo bioavailability. MeHg bioaccessibility from fish muscle has been compared once with oral bioavailability using a mouse model [25]. The authors found no correlation between the two methods. Metals in fish muscle are complexed differently than in soils, which could modify the efficiency of in vitro models to release MeHg from fish muscle. Despite this, recent studies have started to propose changes in consumption guidelines on the basis of in vitro bioaccessibility results alone [26,27]. Furthermore, numerous in vitro studies have reported that the cooking of fish muscle induced a significant reduction of Hg and MeHg bioaccessibility, which suggest a decreased MeHg solubility [26,28,29,30]. If this phenomenon can be confirmed in vivo, current consumption guidelines could be modified.

As the assessment of MeHg bioaccessibility cannot easily be performed in vivo due to ethical constraints [31], studies have used blood total Hg (THg) values as a biomarker of exposure to this heavy metal. Total Hg blood concentrations, a proxy for MeHg levels, are often well correlated with fish consumption and is used as a short-term exposure biomarker [32,33]. In human blood, MeHg is mostly found trapped in red blood cells (RBC) [4]. Arsenic is also found in RBC and serum fractions mostly in the form of AsB [34,35]. Although it is known to be unbound and poorly degraded or metabolized [17], exposure to this organic As species has not been associated with toxic effects because of its high excretion rate [36]. Furthermore, to the best of our knowledge, the effect of cooking on its oral bioavailability has yet to be studied.

Here, we determined the effect of cooking on MeHg exposure after fish consumption, using both in vivo and in vitro approaches. According to literature, cooking should decrease MeHg exposure [26,28,29,30]. In this study, pigs were fed raw or cooked tuna meals, and we collected portal blood samples periodically for 540 min post-meal. Emphasis was put on the serum fraction, since compounds must be distributed in this blood compartment prior to exiting the vascular system [37]. Through this approach, we assessed the kinetics of the net oral bioavailability of MeHg, As, and Se as a function of the cooking method, and we determined the partitioning of MeHg and other metal(loid)s in blood compartments. In parallel, we conducted in vitro digestion using the same tuna meals (raw and cooked) to assess MeHg bioaccessibility. Our research is of interest since other bioaccessibility studies have proposed using in vitro results to modify fish consumption guidelines.

## 2. Materials and Methods

### 2.1. Animals, Surgeries, and Treatments

Five Yorkshire-Landrace × Duroc gilts females were selected at 30 kg of body weight (BW) and fed ad libitum a conventional basal diet for growing pigs (Table 1) until surgery. Average BW at surgery was 45.6 ± 3.9 kg at 80.5 ± 4.9 days of age. The surgery procedure has been described by Hooda et al. [38] and modified by Dalto et al. [39]. Briefly, a catheter was inserted into the portal vein at approximately 2.5 cm before its entry into the liver, and another catheter was inserted through the carotid artery up to the junction between the carotid and subclavian arteries. This is a delicate surgical procedure that was performed by experienced veterinarians following strict ethical guidelines. This imposed a limit to replication with 5 pigs included in this paper, for each of which we collected detailed data over time (see below). This is the first time that this experimental design is used to assess MeHg oral bioavailability.

After surgery, animals were penned individually (1 × 1.8 m) and fed the conventional growing-phase diet described above in a single daily meal in accordance with their BW (1.0 kg/day when <50 kg BW; 1.2 kg/day from 50–60 kg BW; 1.5 kg/day when >60 kg BW). A total of 7 to 10 days after surgery, when the animals had fully recovered (appetite and normal growth rate), they were gradually acclimatized (3–5 days) to the metabolic cage (i.e., a specific type of cage allowing the collection and the separation of urine and feces) with free access to water. One week prior to the experimental day, animals were fed one single portion of raw (80 g) or cooked (80 g) tuna to have them adapt to the taste. Each pig received a single dietary treatment (Table 1) on the experimental day. Exceptionally, Pig 1 was fed a cooked tuna meal (1C) followed by a raw tuna meal (1R) one week later. The other animals received either a raw or cooked treatment only.

For dietary treatments, two yellowfin tuna (*Thunnus albacares*) frozen loins (1.3 to 2.3 kg each) were bought from the Odessa Poissonnier fish market (Montreal, QC, Canada). Once thawed at room temperature, the tuna was chopped into small pieces (1 × 1 cm) and separated into ten meals of approximately 520 g wet weight. The remaining mass was used for the in vitro digestion procedure and was kept at −20 °C until the experimental day.

For the cooked treatment, tuna samples (427 ± 4.0 g) were placed in a previous acid-washed Pyrex™ dish and heated in an oven (VWR Scientific, Randor, PA, USA, 1370 FM forced air oven) until an internal temperature of 70 °C was reached, corresponding to the safe internal cooking temperature according to Health Canada [40]. Raw and cooked tuna samples were kept at −20 °C until the experimental day. Each tuna sample was subsampled (±20 g) and preserved in the freezer until metal(loid) analysis (−20 °C). The final mass of each meal was 495 ± 2.3 g for raw tuna and 375 ± 0.4 g for cooked tuna (due to moisture lost in the oven). Final metal(loid)s levels are presented at Table 1. On average, cooked tuna meals contained 14% more MeHg than the raw treatment, resulting from the heterogenous distribution of MeHg in tuna muscle. Overall, three raw (fed to pigs 1R, 2, and 3) and three cooked meals (fed to pigs 1C, 4, and 5) were administered in the experiment.

### 2.2. Blood Collection and Analysis

On the experimental day, pigs were moved to the metabolic cages, and blood samples (4 mL) were collected simultaneously from the two catheters at five minutes before the dietary treatment, every 30 min for the first two hours after the meal, and every hour for the following seven hours for a total of nine postprandial hours (total blood samples: 12 from the portal vein and 12 from the carotid artery). A total sampling time of 540 min was chosen since mean gastric emptying time and chyme transit time in the small intestine are approximately 3–4 h each, for a total of 8 h (480 min) [41,42,43]. Immediately after sampling, arterial and portal-drained viscera (PDV) blood was transferred through syringes into EDTA-treated tubes (Vacutainer, Becton Dickinson, Franklin Lakes, NJ, USA) and trace element–free BD Hemogard™ Closure tubes (Vacutainer, Becton Dickinson, Franklin Lakes, NJ, USA). Packed cell volume was measured in duplicate on fresh PDV blood by microcentrifugation. Aliquots of collected arterial and PDV blood were frozen for metal(loid) analysis as well as for determining hemoglobin according to the method of Drabkin [44]. After at least four hours of decantation, arterial and PDV serums were then collected after centrifuging the blood samples at 1800× *g* for 10 min at 4 °C. The serums were frozen at −20 °C until used for metal(loid) analysis.

### 2.3. In Vitro Digestion Experiment

We used an in vitro protocol, the physiologically based extraction test (PBET), to test the metal bioaccessibility of raw vs. cooked fish. The PBET was used based on a protocol modified from Ruby et al., Ouédraogo and Amyot, and Girard et al. [28,29,45]. The ratio of gastric and intestinal fluids (1:1) was set according to Smith and Morton [46].

Digestive fluids were prepared on the same day of the experiment in acid-washed Teflon™ bottles. Chemicals were purchased from Sigma-Aldrich (St. Louis, MO, USA). Gastric fluid was composed of 1.25 g of porcine pepsin (>400 U/mg), 0.50 g of sodium citrate (>99%), 0.50 g of malic acid (>99%), 420 μL of lactic acid (>85%), and 500 μL of acetic acid (99.7%), mixed in ultrapure Milli-Q water to a final volume of 1 L. For the intestinal fluid, we mixed 0.60 g of bile salts and 0.15 g of pancreatin (4 × USP grade, lipase > 24 U/mg, protease > 400 U/mg) in a final volume of 250 mL 1M NaHCO_3_. pH targets for the stomach and intestinal fluids were 2 and 7, respectively, and were adjusted using HCl (OmniTrace Ultra, St. Louis, MO, USA, EMD) before the digestion procedure.

Briefly, 1.0 ± 0.1 g of raw or previously cooked tuna was roughly homogenized and added to trace metal–free Falcon™ tubes, and each treatment was performed five times. A total of 22.5 mL of gastric fluid was added to the Falcon tubes, and they were placed in a horizontal incubator (100 rpm) at 37 °C for 1 h. After this step, 22.5 mL of intestinal fluid was added to each tube, and pH was monitored and adjusted if needed before a second horizontal incubation (100 rpm) at 37 °C for 2 h.

Isolation of the gastric and intestinal bioaccessible fraction was performed by a centrifugation step at 3000 *g* for 15 min at 20 °C. Aliquots were then taken and kept at −20 °C until further analysis. Bioaccessibility was calculated as
(1)$$\frac{\mathrm{MeHg}\;\mathrm{in}\;\mathrm{PBET}\;\times \;\mathrm{PBET}\;\mathrm{volume}}{\mathrm{MeHg}\;\mathrm{in}\;\mathrm{fish}\;\times \;\mathrm{fish}\;\mathrm{mass}}\;\times \;100,$$
where MeHg in PBET (ng/L) represents MeHg levels measured in the aliquot collected after the PBET digestive processes, PBET volume (L) is the PBET fluid volume (0.0225 L for the gastric and 0.045 L for the gastrointestinal fractions), MeHg in fish (ng/g) is the MeHg level in the fish samples before PBET digestion, and fish mass (g) is the mass of fish sample used for the PBET simulation.

MeHg mass balance was calculated as follows:(2)$$\frac{\mathrm{mass}\;\mathrm{of}\;\mathrm{MeHg}\;\mathrm{in}\;\mathrm{supernatant}\;+\;\mathrm{mass}\;\mathrm{of}\;\mathrm{MeHg}\;\mathrm{in}\;\mathrm{pellets}\;}{\mathrm{mass}\;\mathrm{of}\;\mathrm{MeHg}\;\mathrm{in}\;\mathrm{tuna}\;\mathrm{flesh}}\;\times \;100,$$
where mass of MeHg in supernatant is the quantity of MeHg that is bioaccessible and mass of MeHg in pellet is the quantity of non-soluble MeHg following centrifugation. We obtained an average of 95 ± 10% (*n* = 18).

### 2.4. Total Mercury Analysis

*Whole blood (WB) and tuna muscle.* Total mercury was measured using a direct mercury analyzer (DMA 80, Milestone Inc., Pittsburgh, PA, USA), in which 100 µL of WB and 0.01 g (dw) of freeze-dried tuna samples (Freezone6, Labconco, Kansas City, MO, USA) were thermally decomposed at 750 °C in an oxygen-rich furnace. Elemental mercury vapors were preconcentrated on a gold amalgamation trap for analysis by atomic absorption spectrophotometry. For WB, conversion from mass to volume was achieved using a density of 1.06 kg/L [47].

*Serum*. Prior to THg analysis in serum, 200 µL was digested in 1500 µL of 70% HNO_3_ (OmniTrace Ultra™, MilliporeSigma, Burlington, MA, USA) for 30 min on a hotplate (95 °C). The digestion was completed using OPTIMA grade H_2_O_2_ by completing the volume up to 2 mL and returning it to the hotplate (95 °C) for 5 h; the volume was then adjusted to 2 mL with Milli-Q water (>18.2 MΩ/cm), diluted 1:4 with 2% HNO_3_. Aliquots were analyzed using CVAFS (Tekran 2600, Tekran Instruments Corporation, Seattle, WA, USA). Whole blood and serum were analyzed by two different protocols. Whole blood is highly viscous, and it was more convenient to use sample mass rather than volume for its analysis.

### 2.5. MeHg Analysis

*WB, serum, tuna muscle, and PBET soluble fraction*. For MeHg analyses, 250 µL of WB, 500 µL of serum, 0.01 g (dw) of freeze-dried tuna samples (Freezone6, Labconco, Kansas City, MO, USA), and 100 µL of PBET soluble fractions were digested overnight at 60 °C in 5 mL of 4M HNO_3_ (Fisher Scientific, Waltham, MA, USA, ACS-pur). Samples were then analyzed with a gas chromatograph coupled to a cold-vapor fluorescence spectrometer (GC-CVAFS) (Tekran 2700, Tekran Instruments Corporation, Seattle, WA, USA) with reference to US EPA Method 1630.

### 2.6. Other Metal(loid)s

*WB and serum.* For other metal(loid)s, we performed the same digestion as for THg serum analysis.

*Tuna muscle*. For other metal(loid)s, freeze-dried tuna samples were digested in 250 μL of HCl (OmniTrace Ultra, EMD, St. Louis, MO, USA) and 250 μL of ultrapure 5% HNO_3_ (OmniTrace Ultra™, MilliporeSigma, St. Louis, MO, USA) for 3 h in a pressure steam sterilizer (#50X 25-quart electric sterilizer, All American, Berlin, Germany). The digestion was completed with 250 μL of ultrapure OPTIMA grade H_2_O_2_; then, ultrapure Milli-Q water (<18.2 MΩ/cm) (EMD Millipore, Burlington, MA, USA) was added to reach a total volume of 10 mL. Samples were analyzed by ICP-MS/MS (8900 Triple Quadrupole, Agilent, Santa Clara, CA, USA).

### 2.7. Arsenic Speciation

*WB and serum.* Arsenic speciation was conducted on subsets of all blood samples. We selected six sampling times (t0, 90, 180, 300, 420, and 540 min) over the concentration-time profile of serum (arterial and venous) and WB (arterial and venous) from pigs 1R and 1C (*n* = 48). Whole blood and serum were treated in the same manner. As detailed by an interlaboratory study on As speciation in whole blood [48], 200 µL was digested in a 5 mL methanol–ultrapure Milli-Q water solution (1:1). Samples were then homogenized using an ultrasonic bath (VWR Model 150 D) for 1 h and were centrifuged at 3000× *g* for 15 min at 20 °C. Finally, the resulting supernatants were filtered through a Captiva EMR-lipid cartridge (Agilent) facilitated by a vacuum system. The water-soluble species in the aqueous phase were then injected into a HPLC-ICP-MS/MS.

*Tuna muscle.* Arsenic speciation was performed on ground freeze-dried tuna muscle following the protocol described in Taleshi et al. [49]. A chloroform–methanol solution (2:1) (>99.8 and 99.9%, respectively, Fisher Scientific, Waltham, MA, USA) was used for the first fractionation. The resulting solution was filtered (Machery-Nagel, Duren, Germany, MN 85/70, 45 mm filter papers). After the evaporation of the filtrate, the oily phase was submitted to a second extraction using chloroform–methanol–ultrapure Milli-Q water solution (2:1:1). The chloroform phase was evaporated, and the oily substance containing fat-soluble As species was digested following the Quebec (Canada) environment ministry (MELCC) protocol MA.–Mét 1.2 (2000) and analyzed by ICP-MS/MS. The water-soluble species in the aqueous phase were then analyzed by HPLC-ICP-MS/MS.

### 2.8. Quality Control for Metal(loid) Analyses

Intercalibration criteria followed that of the Canadian Association for Laboratory Accreditation (CALA). Various certified reference materials (CRM) were used for quality control and were submitted to the same digestion protocol as the samples. For WB and serum, we used SeronormTM Trace Elements Whole Blood L-2 and Serum L-1 certified materials (SERO, Billingstad, Norway) and obtained a mean recovery of 98 ± 0.8% (*n* = 28) for WB THg, 100 ± 3.4% (*n* = 7) for serum THg, 91 ± 3.5% (*n* = 13) for WB and serum MeHg combined, and 90 ± 17.4% (*n* = 34) for WB and serum for other metal(loid)s.

For tuna muscle and the PBET soluble fraction, we used TORT-2 (lobster hepatopancreas, National Research Council, Canada) as well as DORM-2 and DORM-3 (fish protein, National Research Council, Canada) reference materials. Mean recoveries for tuna muscle were 102 ± 2.8% (*n* = 23) for THg, 97 ± 6.5% for MeHg, and 92 ± 0.01% (*n* = 2) for Se and As. Finally, 101 ± 3.4% (*n* = 4) was measured for MeHg in the PBET soluble fraction. Detection limits of metal(loid) analyses are presented in Appendix A.

### 2.9. Data Handling and Statistics

Venous concentrations of metal(loid)s recorded five minutes before the meal were used as basal levels (t0). The fraction (%) of intake metal(loid) in the blood compartment at time t was assessed as [50,51]
(3)$$\frac{M\;\times \mathrm{whole}\;\mathrm{body}\;\mathrm{blood}\;\mathrm{volume}}{\mathrm{intake}}\;\times \;100,$$
where M describes the metal(loid) point concentration, whole body blood volume was estimated at 7% of BW (L/kg) [52,53], and metal(loid) intake corresponds to the quantity of the metal(loid) in tuna meal (μg) (Table 1). Mean pig BW was 56 ± 5.4 kg (CV = 9.6%) on the day of the experiment.

As described and validated by Kershaw et al. [51], RBCs metal(loid) concentrations were calculated as
(4)$$\mathrm{Crbc}=\;\frac{\mathrm{Cwb}-\left(\mathrm{Cserum}\;\times \;\left(1-h\right)\right)}{h},$$
where Crbc, Cwb, and Cserum, correspond to the RBCs, WB, and serum concentrations, respectively, and h is the hematocrit. In the case of negative Crbc, where Cserum > Cwb, Crbc was set to zero.

Kinetic assessment was done using the serum concentration-time profile and exploiting the following pharmacokinetic parameters: peak concentration (C_max_) and time to peak concentration (T_max_) were used as indirect metrics to characterize the rate of absorption, whereas the systemic exposure was assessed through dose-normalized area-under-the-curve from 0 to 540 min post-meal (AUC_0–540_), from the linear trapezoidal method [54,55,56].

Metal(loid) levels as a function of time and treatment were treated with multiple linear regression models and using “individual pig” as a random factor. A post hoc Tukey test was used when appropriate. Due to the non-normality of groups, the mean MeHg profiles from raw and cooked treatments were compared using nonparametric Wilcoxon tests. The comparison of the metal(loid) levels between the portal vein and the carotid artery was done using regression linear analysis when the residuals were normally distributed. Otherwise, the Kendall test was used. Methylmercury bioaccessibility results from raw and cooked treatments were also compared using nonparametric Wilcoxon tests because of the lack of equivariance between groups. The level of significance was set at *p* ≤ 0.05.

## 3. Results

### 3.1. In Vitro Bioaccessibility of MeHg

During in vitro assays, gastric and gastrointestinal (GI: gastric + intestinal bioaccessibility) MeHg bioaccessibility from the cooked treatment differed significantly from the raw treatment, whereas the intestinal bioaccessibility was similar between treatments (Figure 1). We measured a mean gastric and GI bioaccessibility of 41 ± 6% and 64 ± 3%, respectively, for the raw treatment (*n* = 5). It suggests that gastric digestion was mainly responsible for the solubilization of MeHg because the intestinal digestion alone accounted for only 23% of the solubilization. Compared with the raw treatment, a substantial MeHg bioaccessibility decrease was observed using cooked tuna, with bioaccessibility values of 8 ± 1% and 31 ± 2% (*n* = 5) for the gastric and GI compartments, respectively. Again, 23% of the MeHg solubilization occurred in intestinal digestion. These results suggest that cooking decreased consumer exposure to MeHg by 33%.

### 3.2. Effect of Cooking on Methylmercury, Arsenic, and Selenium Kinetics in the Pig Digestive System

During the in vivo experiment, we followed venous serum MeHg concentrations in pigs as a function of time and treatment (Figure 2). In contrast with the in vitro results, MeHg levels from the serum of the raw and cooked treatments were not statistically different (treatment effect, *p* > 0.05). Similar trends were observed for AUC_(0–540)_, with no difference observed between treatments (Table 2). On the other hand, the cooking treatment influenced MeHg levels as a function of time (time effect × treatment, *p* < 0.05), as illustrated by the mean concentrations as a function of time for both treatments (Figure 2, All pigs panel).

The average consumption time for both treatments was 20 min. Increases in MeHg levels in serum (Figure 2) and RBCs (Appendix A) were observed within 30 min. All pigs, with the exception of Pig 2, reached their C_max_ within the recorded 540 min (T_max_ ranged from 120 to 360 min, Table 2). At C_max_, the fraction of total MeHg intake (see Equation (3)) varied from 2.0% to 8.4%, 0.6% to 1.2%, and 1.6% to 7.4% in the WB, serum, and RBC compartments, respectively. These values were not influenced by the cooking treatment (treatment effect, *p* > 0.05). The blood MeHg profile suggests that the declining phase was not finished at 540 min post-meal. Pig 2, fed with raw fish, did not reach C_max_ during the experiment. This pig was healthy the day of the experiment and interindividual variability is our only explanation regarding its MeHg level serum profile (Figure 2).

As mentioned above, treatments influenced MeHg levels in serum as a function of time, and this observation agreed with the observed difference between both treatments in terms of T_max_. Pigs exposed to the cooked meal reached their MeHg C_max_ earlier than those fed with raw meals (Table 2). This treatment effect is supported further by the earlier C_max_ observed for the cooked treatment in Pig 1 (1C) compared with its C_max_ for the raw treatment (1R).

In addition to the treatment responses of the MeHg kinetic profiles, we also observed a generally large interindividual variability. Within the same treatment (raw or cooked), pigs showed markedly different metal(loid) serum profiles as a function of time (Figure 2 and Appendix A). For example, Pig 3 demonstrated MeHg and TAs oral bioavailability profiles that differed markedly from those of Pigs 1R and 2, even though they all received similar doses of both metal(loid)s (Table 2 and Appendix A) (pig effect, *p* < 0.05).

For TAs absorption kinetics, the response was very similar to that of MeHg, including the distinction observed for the Pig 2 profile (Appendix A) and the early absorption recorded in the first 30 min post-meal. As for MeHg, As-serum levels were influenced by treatment as a function of time (time effect × treatment, *p* < 0.05) and AUC_(0-540)_ was similar between treatments (Appendix A); however, T_max_ was not affected by cooking (Appendix A). Before the fish meal, we did not detect any AsB in the blood of any pigs. In tuna meal and postprandial blood, AsB was the only detectable As species. In the pig’s fish meal, AsB represented on average 73 ± 9% and 96 ± 5% of total As in raw and cooked treatments, respectively (Table 1). Average AsB recovery (AsB/TAs × 100) in blood for the raw treatment was 41 ± 16% (range 27–69%, *n* = 23 with one outlier excluded) and 63 ± 11% for the cooked treatment (range 45–95%, *n* = 24). Since AsB was the only As species in tuna meal, it can be used as a specific marker of the absorption of As from tuna meal. Indeed, AsB levels from WB and serum (*n* = 47) were highly correlated to TAs values (*r*^2^ = 0.74, *p* << 0.001). After the meal, AsB absorption profiles were similar to those of TAs, except that AsB concentrations were lower (Appendix A).

Compared with MeHg and TAs, TSe oral bioavailability showed no clear trends (Appendix A). According to the mixed linear model, TSe levels did not differ from zero (time effect, *p* > 0.05) and were not influenced by treatments (time effect × treatment, *p* > 0.05). These results suggest that tuna meal had no impact on the TSe blood levels of pigs.

### 3.3. Stability of Methylmercury, Arsenic, and Selenium Distribution between Blood Compartments

Global venous data were plotted against the arterial data (Figure 3) to evaluate the stability of metal(loid)s within a given blood compartment. In cases where venous and arterial measures tightly fitted the 1:1 line, it would suggest a rapid equilibrium with limited biodistribution of the metal(loid)s between blood compartments (RBCs to serum and vice versa), and from the blood toward extravascular tissues. As illustrated in Figure 3, we obtained a strong correlation for MeHg in the RBC fraction (*r*^2^ = 0.98, *p* << 0.001) for which the linear regression slope (solid line: slope; gray: confidence interval = 0.95) did not differ from the 1:1 slope (dotted line; ANCOVA, *p* = 0.07). Data for MeHg in the serum fraction were not normally distributed (Shapiro test, *p* = 0.07); however, according to the Kendall test, venous and arterial data were well correlated (τ = 0.88, *p* << 0.001). Another strong correlation was observed for As in the serum fraction (*r*^2^ = 0.98, *p* << 0.001), but the regression slope differed from a 1:1 relationship (ANCOVA, *p* << 0.001). In contrast with serum, As in the RBC fraction displayed a significant but weak correlation (*r*^2^ = 0.32, *p* << 0.001), with many data falling beyond the 95% confidence interval. This weaker correlation is unlikely due to analytical constraints because the limit of detection for As was 0.71 ng/L (Appendix A). Finally, for Se in serum (*r*^2^ = 0.86, *p* << 0.001) and RBC (*r*^2^ = 0.70, *p* << 0.001), the results were similar to those of Hg and As, showing a moderate correlation.

### 3.4. Cooking Effect on Metal(loid) Distribution between Blood Compartments

We investigated further whether cooking affects MeHg blood partitioning. Figure 4 shows the proportion of metal(loid)s when the blood compartment volume was considered. RBCs constitute the main reservoir of MeHg and THg, representing, respectively, 81 ± 4% (*n* = 34) and 84 ± 5% (*n* = 35) of the total load in the blood of pigs fed raw meat and 86 ± 3% and 79 ± 13% (*n* = 36 for both), respectively, in pigs fed cooked tuna meal. Contrasting results were found for Se and As. Indeed, they were not preferentially distributed in the RBC fraction, with 38 ± 4% and 45 ± 7% for Se and 9 ± 7% and 15 ± 9% for As in pigs fed raw and cooked tuna, respectively. For THg, Se, and As, but not for MeHg (Figure 4), the distribution between blood fractions varied as a function of time, without any clear trend, as for other elements (Appendix A) (time effect, *p* < 0.05). For all studied metal(loid)s, these proportions did not differ statistically between treatments (treatment effect, *p* > 0.05) (Figure 4 and Appendix A). When comparing concentrations instead of loads, RBCs contained 11 to 15 times more MeHg, 2 times more Se, and 3 times less As than the serum (Appendix A).

## 4. Discussion

### 4.1. Differing Effects of Cooking on the In Vitro and In Vivo Model Results

Overall, cooking decreased in vitro MeHg bioaccessibility by a factor of 2, with a GI bioaccessibility of 64% and 31% for the raw and cooked tuna, respectively (Figure 1). Other digestion models have similarly established that cooking decreases Hg and MeHg solubility within GI fluids [57,58,59,60]. A decrease in fish MeHg bioaccessibility following heat treatment is usually explained by protein oxidation produced by the heat treatment through various cooking methods, leading to the modification of the amino acids [61,62]. Ultimately, protein polymerization and aggregation occur [63], which limits enzymatic protein degradation and digestibility [64]. Because MeHg is bound to protein [65], cooking could decrease the MeHg release from fish muscle into GI fluids.

The effect of cooking on bioaccessibility likely results from processes occurring during the gastric phase, as no difference was observed for the intestinal phase. Indeed, we observed five times lower gastric MeHg bioaccessibility for the cooked treatments, which can be linked to the reduced efficacy of the gastric pepsin with the digestion of cooked fish muscle. Concordant results have been found for the in vitro cooking effect on myofibrillar protein digestibility [63]. In that study, myofibrillar proteins were isolated from bovine meat (these proteins are also present in fish muscle; [66]). After 5 min of cooking at 100 °C in a dry bath, the proteolysis rate for gastric pepsin decreased by 42% relative to the control [63], suggesting a non-negligible effect of cooking on in vitro gastric protein digestibility.

In contrast to these above-mentioned studies, we found that cooking did not affect the postprandial MeHg concentrations in the serum and RBCs of pigs fed with fish. The principal observed effect in the in vivo experiment was a faster uptake in the cooked treatment. Similar in vivo reports are consistent with our observations. A study using ileostomized humans, in which the aim was to evaluate the effect of cooking temperature in a steam oven on ^15^N-labeled bovine meat digestibility—assessed by the appearance of ^15^N-labeled in the ileum—found no statistical difference between barely cooked meat (55 °C for 5 min, *n* = 8) and highly cooked meat (90 °C for 30 min, *n* = 8) [67]. Comparing this human-based study to our pig-based one is relevant in the context where a pig’s digestive system is comparable to the human one [24]. Even from this perspective, to our knowledge, there is no other relevant literature addressing the effects of cooking on in vivo digestibility/absorption of metal(loid)s.

According to the in vivo results, cooking does not significantly modify MeHg oral bioavailability. We could explain our results by a potential intestinal membrane competition between MeHg and amino acids released from fish muscle. Indeed, in fish muscle, MeHg is believed to bind predominantly to cysteine [68], and the MeHg-Cys complexes can cross the intestinal membrane via B^0,+^ amino acid transporters because of the molecular mimicry of the amino acid methionine [69,70]. Moreover, Vazquez et al. found that MeHg transport across the intestinal barrier is a saturable mechanism [71]. That said, a higher level of free amino acids in the intestinal lumen may reduce the absorption of MeHg by means of transport competition [72]. As we observed a higher in vitro MeHg solubility and a potentially higher solubilization of amino acids in the raw treatment, this could have limited intestinal MeHg absorption, leading to a similar MeHg oral bioavailability in pig blood. Our study suggests that not all the soluble MeHg is absorbed by the intestinal barrier. We have not found any studies focused on the impact of cooking on the MeHg-Cys complex.

Together, our results suggest that the MeHg in vitro bioaccessibility model used in this study is not a direct surrogate for in vivo models. In vitro models remain useful complementary tools to test different gastric and intestinal processes affecting metal bioavailability from diet under controlled conditions. They also have the advantage of being non-invasive and highly replicable, but are not optimized to directly predict mammalian MeHg exposure. To the best of our knowledge, this is the first direct comparison of raw and cooked tuna meat using in vitro bioaccessibility combined to in vivo bioavailability approaches for MeHg.

### 4.2. Cooking Does Not Influence the MeHg Fraction in Blood

Blood is a short-term Hg exposure biomarker of choice because Hg is slowly excreted from organisms [73]. Maximal MeHg blood levels (C_max_ without subtracting the concentration from t0) observed in our pig WB were considerably under the acceptable Canadian general population guidance (<20 µg/L) [74], varying from 1.1 to 3.8 μg/L. The fraction of total MeHg intake in the WB compartment remained low. At its highest proportions, the MeHg fraction varied from 2.0% to 8.4% (see Equation (3)) and was not affected by cooking. This range or MeHg fraction is generally consistent with what has been reported for other species. For instance, at a steady state, the MeHg body burden found in human WB for persons having been exposed through fish consumption is estimated at 5.6% to 5.9% [51,75]. Values of 5.5% were found in common loon (*Gavia immer*) chicks (35 days of oral or intravenous exposure) and 10.4% in Cory’s shearwaters (*Calonectris diomedea*) (multiple months of oral exposure), both exposed to MeHg chloride [51,76]. The wide range of values observed in our pigs, relative to previously published values, could be due to interspecific variation in terms of MeHg blood affinity and stability [77], in addition to MeHg not being at a steady state.

### 4.3. Cooking Affects Methylmercury and Arsenic Kinetics of Oral Bioavailability in Blood

Cooking did not change MeHg levels in blood, relative to levels in the blood from the raw tuna test; we nonetheless investigated further the potential effect of heat treatment on MeHg kinetics of oral bioavailability. Knowing that cooking can modify protein configuration, it may have an effect on its digestibility rate.

Cooking affected MeHg levels as a function of time (time effect × treatment, *p* < 0.05). This observation was also supported by the T_max_ results (Table 2), which demonstrated an earlier C_max_ for pigs fed with cooked tuna meal. This phenomenon was also observed in Pig 1, which received both types of treatments (1R and 1C), suggesting that the observed variance would not only be related to interindividual variability but also due to meal properties. Studies on ileostomized humans illustrate a faster ^15^N-labeled uptake in ileum for a highly cooked bovine meat than for a minimally cooked meat [67]. Changes in tuna meal viscosity by cooking treatment could be responsible for the observed T_max_. Indeed, gastric emptying time is negatively correlated with meal viscosity [78], and cooking decreases the viscosity [79]. The earlier T_max_ observed for the cooked meal could therefore be related to a lower meal viscosity and a faster gastric emptying, relative to the raw treatment.

Mean gastric emptying time and chyme transit time in the small intestine are 3–4 h each, resulting in 8 h [41,42,43]. Consequently, our experimental design was set at 9 h (540 min) of blood sampling, a time period deemed sufficient to record the full MeHg bioavailability profile. However, according to Figure 2, a longer monitoring period would have been preferable, although difficult to perform under the logistical constraints of the invasive surgery [51].

In our study, AsB was the only As species detected in tuna meal and blood. This As species is known to be readily absorbed [80]; however, to the best of our knowledge, the effect of cooking on its absorption has yet to be studied. The kinetics of oral bioavailability for As in serum were altered by cooking. Because the AsB profile was similar to that of TAs, and AsB values correlated well with TAs, it is likely that cooking would also influence the AsB kinetics of oral bioavailability. Lehmann et al. studying blood AsB levels after a single fish meal (cooked plaice filet) in 14 women, observed a distribution half-life of 7.1 h and elimination half-life of 63 h [81]. This agrees well with a previous study where human volunteers (*n* = 15) were fed with a single meal of cooked flounder. After eight days, the subjects had 76% of the AsB doses in urine and 0.33% in feces [80]. The authors suggest that the small proportion of AsB in the feces relates to AsB being readily absorbed and excreted without any metabolic change. This fast elimination rate could be related to the As distribution found mostly in the serum fraction (Figure 4), thereby implying a higher bioavailability for a biodistribution toward the elimination pathway.

Tuna meal did not alter TSe serum levels, as the values did not differ from zero (time effect, *p* > 0.05) (Appendix A). Being an essential element [82] that is normally well absorbed [83], our results could be related to the tight homeostasis and intense metabolic Se requirements for intestinal tissue as proposed by Dalto and Matte [84]. Furthermore, once absorbed, this metalloid may be incorporated into essential selenoproteins [85], such as selenoprotein P (SelP) which acts as a major reservoir of Se, stored in the liver [85]. Hence, it was anticipated that such levels of Se from single tuna meals would not significantly alter Se levels in serum.

### 4.4. Partitioning of MeHg in Blood Compartments Differed for As and Se

Tracing the MeHg distribution ratio between RBCs and serum is important because it could influence the distribution of MeHg in tissues and its toxicity [37]. To exit the vascular system, MeHg must be distributed in the serum fraction [86], which may result in a delay. The European Food Safety Authority has proposed that the delay of the entry of MeHg into the brain could be the result of the binding of MeHg to RBCs [4].

Methylmercury was chemically stable over time in both blood compartments and was largely distributed in the RBC fraction. As well, MeHg distribution between both compartments was not affected by the cooking of tuna (Figure 4). Since no study has assessed the effect of cooking on MeHg complexes in fish muscle, gastrointestinal juices and blood of consumers, this is a relevant result suggesting that speciation of MeHg from cooked or raw muscle is similar, resulting in a comparable metabolism when absorbed.

In blood, MeHg can form complexes mainly with sulfhydryl group–rich (-SH) biomolecules, such as hemoglobin, albumin, cysteine, and glutathione [77,87], as well as with selenol (-SeH) [88]. The mobility of MeHg is attributed to the rapid exchange reaction between MeHg-bound ligands [89]. An in vitro study using human RBCs demonstrated that the exchange of MeHg between glutathione molecules is less than 0.01 s [90]. However, this study used particularly high MeHg levels, which could have influenced their findings [89]. In bottlenose dolphins (*Tursiops truncatus*), RBCs had greater concentrations of the -SH group than in plasma (11,730 ± 587 µM vs. 425 ± 3 µM, respectively) [91], likely because hemoglobin found in RBCs is thiol-rich [88]. Indeed, in the bottlenose dolphin study, the authors observed a direct relationship between -SH group and MeHg bioaccumulation in blood compartments, suggesting that an asymmetric -SH distribution between the plasma and RBCs is responsible for the MeHg distribution [91].

Furthermore, the 1:1 relationship between the venous and arterial RBC values implies that MeHg levels measured immediately after intestinal absorption are very similar to those measured after the blood had passed through the hepatic, pulmonary, and cardiac systems. This suggests a slow biodistribution outside of the circulatory system and a low bioavailability for the surrounding tissues [86]. These observations are consistent with the observed delays between food consumption and biodistribution in humans [4].

Arsenic and Se were likely more mobile than MeHg in the blood system (Figure 3). This difference could explain the observed variation of these metalloid proportions found in the RBC fraction as a function of time in Figure 4 (time effect, *p* < 0.05). Specifically, with respect to As in the RBC venous concentrations were generally greater than the arterial levels, suggesting potential As exchange from RBCs to the serum compartment and subsequent extravascular biodistribution between the portal vein and the carotid artery. It cannot be ruled out, however, that As from dietary treatments would be absorbed continuously by the intestinal wall during the digestive processes that lasts for approximately four hours. This dietary input would possibly result in higher As levels in the portal vein compared with the carotid artery.

In our study, AsB was the only As species detected in tuna meal (Table 1) and postprandial blood, and it was preferentially found in the serum fraction (Appendix A). AsB in human serum has been found unbound, such as its structural analog glycine betaine [92] and is very stable and not inclined to be degraded or metabolized [17]. It would, therefore, be unlikely that AsB would decrease the potential antagonistic effect of Se toward MeHg in blood. Metabolism of that As species in blood remains unknown [17].

For Se, Figure 3 suggests a biodistribution of Se toward extravascular tissue or exchange between blood compartments. Selenium partitioning between blood compartments depends highly upon its speciation. In human plasma, three major species have been identified: SelP, glutathion peroxidase, and the selenoalbumin complexes [93]. Achouba et al. found that in plasma, SelP accounted for 52% of total Se and that up to 50% of Hg was associated with SelP [94]. Another study proposed the formation of the MeHg-Se-SelP complex in rat plasma, which reduced Hg plasma bioavailability and facilitated Hg excretion [95]. In RBCs, Se may be found as selenoneine, known to bind to hemoglobin and prevent Fe oxidation [93]. RBCs collected from Nunavik Inuit, a Canadian population exposed to high levels of selenoneine through beluga consumption, up to 92% of Se was found to be selenoneine (mean of 26%, *n* = 858) [93]. Another study found that selenoneine was the dominant Se species in pig kidneys as well as in the blood and skeletal muscle of tuna [96]. Selenoneine is of interest in studies of MeHg because selenoneine can promote MeHg demethylation and excretion in zebra fish embryos [97]. Therefore, MeHg in the RBC fraction of pig’s blood could be bound to the selenoneine from tuna muscle and, as speculated by Achouba et al., could convert MeHg into an insoluble (nontoxic) mercury selenide (HgSe) and contribute to its excretion [98].

## 5. Conclusions

Our study demonstrates that MeHg bioaccessibility based on in vitro models should be used as a complementary tool to in vivo studies and not as a substitute. While previous studies suggested using direct bioaccessibility results in risk assessment calculation, our results highlight that in vitro digestion model may lack representativeness. The direct use of in vitro bioaccessibility model to inform risk assessment of Hg exposure from fish consumption should be revisited. Indeed, our results suggest that not all of the soluble, bioaccessible fraction of MeHg will be further absorbed by the intestinal wall and that potential competition during membrane transport could occur with amino acids. To strengthen this hypothesis, however, more research is required that focuses on the binding of metal after cooking.

To our knowledge, ours is the first study to assess the impact of cooking on MeHg, As, and Se nutritional metabolism and distribution in blood using the pigs as an experimental model for humans. Our results show that MeHg from cooked tuna meal was absorbed faster than raw tuna, despite the observed interindividual variability. Once in the blood compartment, MeHg was stable for at least 540 min postprandial, in contrast to As and Se for which their distribution between blood fractions varied as a function of time.

Finally, more in-depth evaluation of the interactions between MeHg, Se and As should be conducted in blood and other tissues where these elements bioaccumulate, following a long-term exposure of raw and cooked fish meal. These studies could rely on metallomic tools such as liquid chromatography coupled to inductively coupled plasma mass spectrometry [99]. This would increase our knowledge towards the mechanism of detoxication of MeHg coming from raw and cooked fish meal and see if cooking affects the behavior and the toxicity of MeHg.

## Figures and Tables

**Figure 1 toxics-09-00027-f001:**
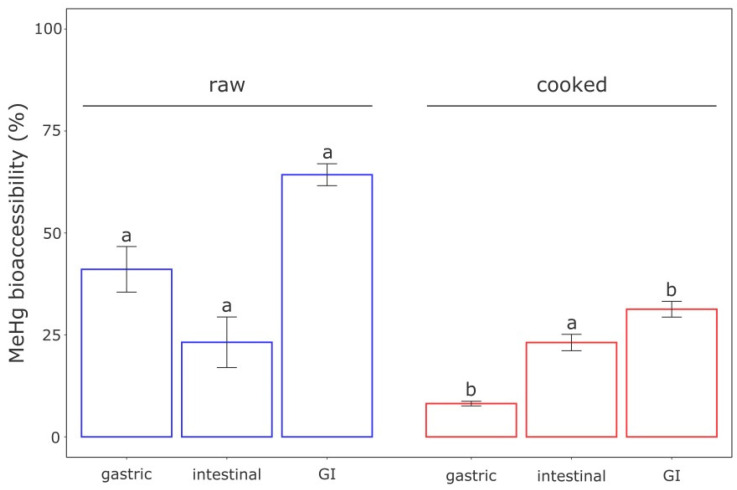
Methylmercury bioaccessibility (%) according to the digestive compartment assessed from raw and cooked tuna muscle (*n* = 5). GI: gastric plus intestinal bioaccessibility. Letters indicate a significant difference (Wilcoxon tests, *p* ≤ 0.05) between treatments. Error bars present the standard deviation of the quintuplicates.

**Figure 2 toxics-09-00027-f002:**
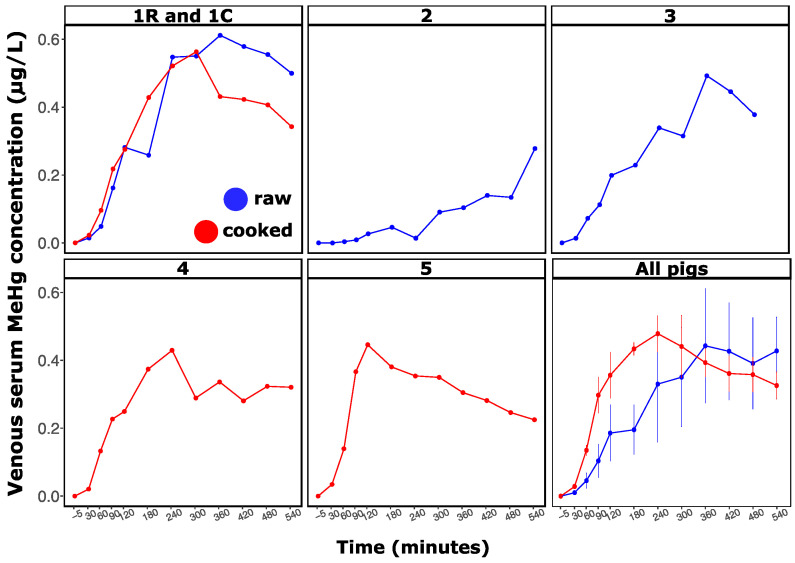
Venous serum MeHg concentration (µg/L) of five individuals during the 540 min post-consumption of a single raw (1R, 2, 3) or cooked tuna meal (1C, 4, 5). Values differed from zero (time effect, *p* < 0.05) and were influenced by treatments (time effect × treatment, *p* < 0.05). Panel “All pigs” presents the average (± SEM) of venous serum MeHg concentration (µg/L) for raw (*n* = 3) and cooked (*n* = 3) tuna meals. Serum MeHg averages from raw and cooked treatments were similar (Wilcoxon test, *p* > 0.05). Preconsumption concentration was subtracted from each value.

**Figure 3 toxics-09-00027-f003:**
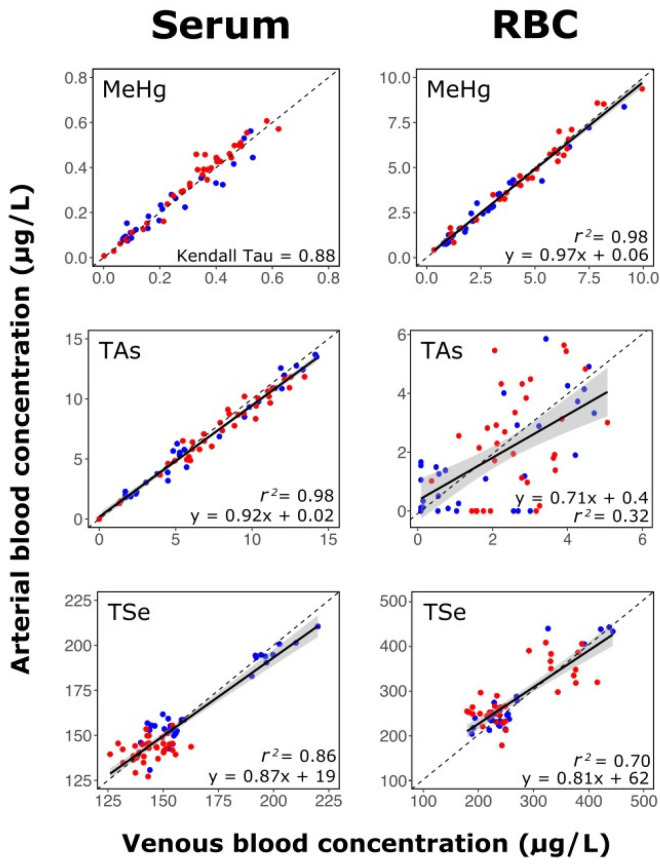
Relationship between the concentration of metal(loid)s in venous and arterial blood for all pigs and sampling times (*n* = 72). Blue points represent raw tuna meal, and red points represent cooked tuna meal. The dotted line shows the 1:1 slope, and the solid line signifies a significant regression slope with the gray area illustrating the confidence interval at 0.95.

**Figure 4 toxics-09-00027-f004:**
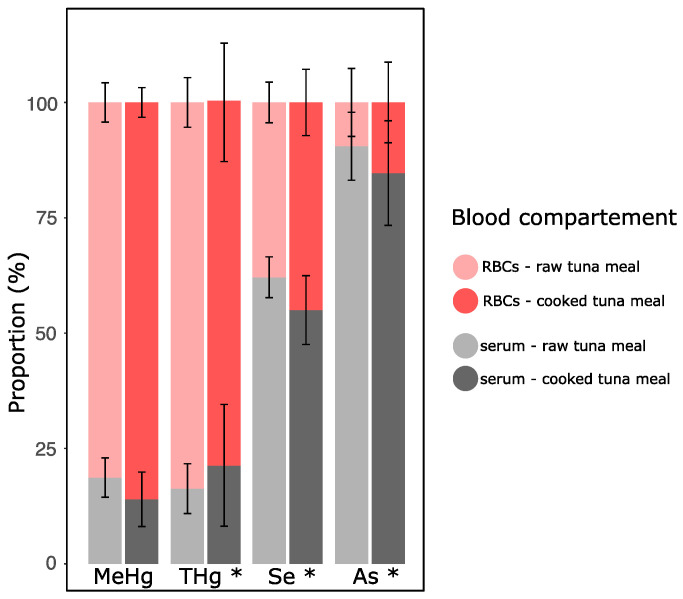
Mean (± SD) proportion (%) of the total quantity of different elements distributed bebetween RBCs and the serum fraction over 540. min postprandial (*n* = 34 and 35 for the raw tuna meal, and *n* = 36 for the cooked tuna meal treatment). Total quantity of each blood compartment was adjusted to their respective volumes (RBC concentration was adjusted as a function of hematocrit (M_RBC_ concentration × blood volume × hematocrit) and serum concentration as a function of serum volume (M_serum_ concentration × blood volume × (1−hematocrit)). * indicates that mean proportion varies as a function of time (time effect, *p* < 0.05). No difference was measured between the raw and cooked treatments (treatment effect, *p* > 0.05).

**Table 1 toxics-09-00027-t001:** Conventional basal diet for growing pigs and treatments. Metal(loid)s values (wet weight) are presented as average ± standard error.

Basal Diet, Nutrients ^1.^						Amount (%)
Corn						55.4
Soybean meal 48%						15.1
Wheat						15.0
Barley						1.3
Distillers dried grain with solubles						9.8
Amino acids						3.5
**Basal Diet, Contaminants ^2^ (*n* = 5)**						**Levels (ng/g)**
THg						2.6 ± 1.3
TSe						58.4 ± 7.5
TAs						ND
**Treatment ^3^ (*n* = 6)**	**Pig Number**	**MeHg (µ** **g)**	**%MeHg ^4^**	**TSe (µ** **g)**	**TAs (µ** **g)**	**%AsB ^5^**
Raw	1R *, 2, 3	118.8 ± 26.7	94 ± 8%	319.2 ± 39.4	508.8 ± 16.3	73 ± 9%
Cooked	1C *, 4, 5	139.2 ± 9.4	93 ± 3%	409.4 ± 45.4	598.4 ± 83	96 ± 5%

^1^ Basal diet contained 17.8% protein, 0.73% Ca, 0.52% P, 140 ppm Zn, 18 ppm Cu, 479 ppm Fe, 143 ppm Mn (analyzed values), and 3082 kcal (calculated value). ^2^ THg for total Hg; TSe for total Se; TAs for total As. ^3^ Raw tuna meal corresponded to 495 ± 2.3 g and cooked tuna meal to 375 ± 0.4 g. ND: not detected. * Pig 1 received both cooked and raw treatment a week apart. ^4^ %MeHg is the proportion of THg that is methylated. ^5^ %AsB si the proportion of As that is AsB.

**Table 2 toxics-09-00027-t002:** Details of the dose of MeHg ingested and MeHg changes in pig serum.

	Raw			Cooked		
Pig Number	1R *	2	3	1C *	4	5
**Preconsumption values (μg/L)**	0.20	0.07	0.09	0.06	0.08	0.002
**AUC_(0–540)_ (μ** **g × min/L/μ** **g)**	1.7	0.3	1.8	1.5	1.2	1.1
**Cmax (t0 corrected) (μg/L)**	0.6	0.3	0.5	0.6	0.4	0.5
**Tmax (min)**	360	NA ^†^	360	300	240	120
**Intake (μg)**	130	138	88	137	149	131
**Dose (μg/kg bw)**	2.5	2.4	1.7	2.6	2.3	2.3

* Pig 1 received both cooked and raw treatment a week apart. ^†^ Pig did not reach Tmax.

## Data Availability

The data presented in this study are available in the Appendix A.

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
