# Peer review of "Assessment of In Vitro Bioaccessibility and In Vivo Oral Bioavailability as Complementary Tools to Better Understand the Effect of Cooking on Methylmercury, Arsenic, and Selenium in Tuna"

_toxics, 2021, doi:10.3390/toxics9020027_

Round 1

Reviewer 1 Report

The article entitled Assessment of in vitro bioaccessibility and in vivo oral bioavailability as complementary tools to better understand the effect of cooking on methylmercury, arsenic, and selenium behavior is a very good scientific work showing the influence of thermal treatment on bioavailability of MeHg.

The main finding:
"Our study demonstrates that MeHg bioaccessibility based on in vitro models should be used as a complementary tool to in vivo studies and not as a substitute"
is crucial and shows that the conclusions of in vitro studies cannot be simply transferred to living organisms.

The study also contains an important novelty item because this is the first study to assess the impact of cooking on MeHg, As, and Se nutritional metabolism and distribution in blood using the pigs as an experimental model for humans.

In addition, the article is clearly written. The studies are planned and performed correctly, as well as conclusions from the obtained resuslts.

In my opinion, the article is very valuable and in its current form can be accepted for publication in Toxics.

Author Response

The article entitled Assessment of in vitro bioaccessibility and in vivo oral bioavailability as complementary tools to better understand the effect of cooking on methylmercury, arsenic, and selenium behavior is a very good scientific work showing the influence of thermal treatment on bioavailability of MeHg.

The main finding:

"Our study demonstrates that MeHg bioaccessibility based on in vitro models should be used as a complementary tool to in vivo studies and not as a substitute" is crucial and shows that the conclusions of in vitro studies cannot be simply transferred to living organisms.

The study also contains an important novelty item because this is the first study to assess the impact of cooking on MeHg, As, and Se nutritional metabolism and distribution in blood using the pigs as an experimental model for humans.

In addition, the article is clearly written. The studies are planned and performed correctly, as well as conclusions from the obtained resuslts.

In my opinion, the article is very valuable and in its current form can be accepted for publication in Toxics.

The authors thank the reviewer for the comments.

Reviewer 2 Report

Abstract

Line 22: When I first read catharized I originally thought you were collecting urine. It wasn’t until the methods that I realized that it was only blood. Please reword the sentence to be clearer e.g. We fed pigs raw and cooked tuna meals and collected blood samples from catheters in the portal vein and artery carotid for 540 min post-meal.

The time interval that blood was collected at would or a number of samples taken also be good to add to the abstract.

Introduction

Line 35: this sentence is difficult to understand: maybe change “such as at” to “for example” also change levels to concentrations

Line 36: if concentrations that are known to be toxic to these systems are ingested, they are not becoming toxic, but rather already possess a risk for toxicity.

Line 40: flesh is not a commonly used term when describing fish tissue- do you mean muscle and/ or skin? Please replace throughout the article

Line 46: there is more than one inorganic form of As. If you mean both please change to form forms and if only one form interacts with As please state the form.

Line 58: Put “The”  or “A” before pig model

Line 59: Please add a sentence on why can the pig model be used as a surrogate for humans?

Line 77: Above you mention that Inorganic As interacts with Se, but only discuss AsB in the blood. Is inorganic As not found in the blood?

Lines 82-88: This read more like methods/results. Please rewrite this paragraph to state the objectives, hypotheses, and importance.

Methods:

Lines 90-101: Where any ethics obtained? Please include a statement about where ethics were obtained from

Lines 111-112: What is a metabolic cage?

Table 1/ lines 114-116:  You reference Table 1 and then a treatment in the following sentence but this treatment 1C is not in the table. Either modify table 1 to include the treatments or include in-text. This information needs to be explicitly stated prior to the 1C reference being used.

Lines 130-131. You reference pigs 1R, 2, and 3. These are treatments to pigs, not the pigs themselves.

How was a time duration of 540 minutes chosen and was this enough time? Also please a sentence in the discussion about the appropriateness of time.

Lines 269: Equivariance is tested between groups, not datasets.

Section: 2.3 Again it would help to list out the treatments.

Results:

Fig 2: Six individual makes it seem like there were 6 pigs. Six treatments in 5 pigs is more accurate. This needs to be fixed throughout the paper- there are multiple instances

Table 2: In the caption state that the NA is because the pig did not reach Tmax. Tables should be stand-alone and not need accompanying text. You could also add an asterix and a footnote at the bottom denoting that pig was different from the rest.

I think Fig S2 is much more compelling at demonstrating your point. I would put this in the main body of the paper, but this is just a suggestion. If you arranged your figure into pigs, one having two treatments that would leave a 6th for the combined plot and you could keep a 6 panel.

Figure 3: for all pigs and time points?

Figure 3/3.3 It is unclear why you used different methods to analyze the different relationships. I understand that your data was (weakly non-normal) but that doesn’t preclude you from using a linear regression- regression assumptions are on the residuals. All the data for this section should be analyzed in the same way to make it comparable.

Discussion

Line 411: serum and whole blood?  Or maybe just say blood.

Lines 457-458: interspecific. Shouldn’t this be interindividual?

Section 4.2 and 4.3: The tuna was cooked without oil. How representative is that of cooking practices of fish and could the use of oil alter the results of this study?

Section 4.4: Again, there is no discussion related to my comment in the introduction related to the different species of As in relation to risk. Line 532 mentions that only AsB was detected in the tuna meal. It would be good to have a summary table of what metals and species were measured and the average concentration. Further on this point, at the beginning of the article, it seems like you may investigate the potential interaction between As, Se, and MeHg but when I got to the end of the discussion it doesn’t seem like this was accomplished. I would suggest either rewording the beginning or adding a paragraph to the discussion to accomplish this.

Discussion/Conclusion: In the introduction, you state that this will be useful for risk assessment, but the discussion lacked the echo of this, as risk assessment wasn’t really discussed and as a result, this paper is missing the “so what” component for risk assessment. Please add a few sentences addressing this.

Author Response

Abstract

Line 22: When I first read catharized I originally thought you were collecting urine. It wasn’t until the methods that I realized that it was only blood. Please reword the sentence to be clearer e.g. We fed pigs raw and cooked tuna meals and collected blood samples from catheters in the portal vein and artery carotid for 540 min post-meal.

The authors thank the reviewer for the comment. Change has been done. (L22-24)

The time interval that blood was collected at would or a number of samples taken also be good to add to the abstract.

Done.

Introduction

Line 35: this sentence is difficult to understand: maybe change “such as at” to “for example” also change levels to concentrations

Done.

Line 36: if concentrations that are known to be toxic to these systems are ingested, they are not becoming toxic, but rather already possess a risk for toxicity.

We have changed the verb “become” for “be”. (L36)

Line 40: flesh is not a commonly used term when describing fish tissue- do you mean muscle and/ or skin? Please replace throughout the article

The authors changed “fish flesh” for “fish muscle”.

Line 46: there is more than one inorganic form of As. If you mean both please change to form forms and if only one form interacts with As please state the form.

More than one inorganic As species can interact with Se. Form has been changed for “forms” (L46)

Line 58: Put “The”  or “A” before pig model

“The” has been added. (L59)

Line 59: Please add a sentence on why can the pig model be used as a surrogate for humans?

Clarification has been made. (L58-60)

Line 77: Above you mention that Inorganic As interacts with Se, but only discuss AsB in the blood. Is inorganic As not found in the blood?

Yes, inorganic As could be found in blood but in very small proportion compared to AsB.

The authors have added “mostly” for clarity.

L76: Arsenic is also found in RBC and serum fractions mostly in the form of AsB [34], [35].

Lines 82-88: This read more like methods/results. Please rewrite this paragraph to state the objectives, hypotheses, and importance.

L81-92: modifications has been made and highlighted in yellow.

Methods:

Lines 90-101: Where any ethics obtained? Please include a statement about where ethics were obtained from

L95-105: The section “2.1. Funding sources & approval for animal research” has been added.

Lines 111-112: What is a metabolic cage?

It is a specific type of cage allowing the collection and the separation of urine and feces. It has been added at lines L131-132.

Table 1/ lines 114-116:  You reference Table 1 and then a treatment in the following sentence but this treatment 1C is not in the table. Either modify table 1 to include the treatments or include in-text. This information needs to be explicitly stated prior to the 1C reference being used.

Table 1 has been modified.

Lines 130-131. You reference pigs 1R, 2, and 3. These are treatments to pigs, not the pigs themselves.

The authors understand the confusion. the word “fed” has been added for clarity.   

L152-153: Overall, three raw (fed to pigs 1R, 2, and 3) and three cooked meals (fed to pigs 1C, 4, and 5) were used in the experiment.

How was a time duration of 540 minutes chosen and was this enough time? Also please a sentence in the discussion about the appropriateness of time.

Mean gastric emptying time and the transit of chyme in the small intestine are 3-4h each, resulting in 8 hours. We therefore considered that 9 hours (540 min) of blood sampling would be sufficient. See L159-161 and L515-520

Lines 269: Equivariance is tested between groups, not datasets.

L297: The authors agree. “data set” has been changed for “groups”

Section: 2.3 Again it would help to list out the treatments.

Modification has been made at L172-173

Results:

Fig 2: Six individual makes it seem like there were 6 pigs. Six treatments in 5 pigs is more accurate. This needs to be fixed throughout the paper- there are multiple instances

The authors agree. Changes have been made in every figure caption when needed (main manuscript and SI).

Table 2: In the caption state that the NA is because the pig did not reach Tmax. Tables should be stand-alone and not need accompanying text. You could also add an asterix and a footnote at the bottom denoting that pig was different from the rest.

The authors agree. Modification has been made (L337)

I think Fig S2 is much more compelling at demonstrating your point. I would put this in the main body of the paper, but this is just a suggestion. If you arranged your figure into pigs, one having two treatments that would leave a 6th for the combined plot and you could keep a 6 panel.

Changes have been done. See L325-327 and Fig. 1

Figure 3: for all pigs and time points?

Sampling times have been added in the caption. “Relationship between the concentration of metal(loid)s in venous and arterial blood for all pigs and sampling times (n = 72).” (L400)

Figure 3/3.3 It is unclear why you used different methods to analyze the different relationships. I understand that your data was (weakly non-normal) but that doesn’t preclude you from using a linear regression- regression assumptions are on the residuals. All the data for this section should be analyzed in the same way to make it comparable.

The residuals were not normally distributed precluding us from using this approach. We have added this information about residuals non-normality in the methods (L293-295)

Discussion

Line 411: serum and whole blood?  Or maybe just say blood.

Cooking treatment did not affect MeHg concentration either in serum or RBC. Specification has been made at L448

Lines 457-458: interspecific. Shouldn’t this be interindividual?

No. The authors compared the pig values with literature based on various animal species. (L493-496)

Section 4.2 and 4.3: The tuna was cooked without oil. How representative is that of cooking practices of fish and could the use of oil alter the results of this study?

It is true that people usually add at least a bit of olive oil or butter when baking fish in the oven. Our goal here was not to replicate a specific recipe but to test the general impact of heating/cooking the food. We have already shown elsewhere that the addition of other food item can modify bioaccessibility,(see Girard et al., 2018), so we chose to avoid any interference caused by the addition of such items. Note that in the specific case of oil, experiments conducted by Afonso et al. (2015) did not reveal an impact of oil on MeHg bioaccessibility.

See Afonso et al., “Benefits and risks associated with consumption of raw, cooked, and canned tuna (Thunnus spp.) based on the bioaccessibility of selenium and methylmercury,” Environ. Res., vol. 143 Part B, no. November 2015, pp. 130–137, 2015.

Section 4.4: Again, there is no discussion related to my comment in the introduction related to the different species of As in relation to risk. Line 532 mentions that only AsB was detected in the tuna meal. It would be good to have a summary table of what metals and species were measured and the average concentration. Further on this point, at the beginning of the article, it seems like you may investigate the potential interaction between As, Se, and MeHg but when I got to the end of the discussion it doesn’t seem like this was accomplished. I would suggest either rewording the beginning or adding a paragraph to the discussion to accomplish this.

Table 1 has been modified. The absence of potential antagonism between AsB and Se is discussed at lines L583-588.

Discussion/Conclusion: In the introduction, you state that this will be useful for risk assessment, but the discussion lacked the echo of this, as risk assessment wasn’t really discussed and as a result, this paper is missing the “so what” component for risk assessment. Please add a few sentences addressing this.

The authors added more information on the impact of our results on risk assessment at L608-612.

Reviewer 3 Report

This is a relevant work providing much necessary in vivo data on the effect of cooking over MeHg bioavailability, since up-to-date most works were performed in vitro. In spite the low number of animals used authors compared animal data with in vitro experiments and clearly point the differences in bioavailability between the two different set-ups.

I have some comments/questions:

- From table 1 it seems cooked meals had higher concentration of MeHg, which is normal due to water loss. Why didn’t the authors adjust the mass given to pigs for the cooked meal to guarantee that they were getting the same amount of MeHg than the pigs that were fed the raw meal? Since the meal mass was the same between raw and cooked, the pigs getting cooked got more MeHg than the pigs getting raw meal.

- Why wasn’t total Hg in serum analyzed by the same method as Hg in total blood? Was it related with the sample amount and the need to analyze also Se and As? Please clarify.

- Lines 472-475: There seems to be a confusion here. If cooking decreases viscosity and therefore gastric empty is faster, the two variables are negatively correlated (as on decreases the other increases) and not positively as written

Lines 491 – 494: Keep in mind the absorbed Se (regardless it is inorganic or organic) does not go free into circulation. It is largely integrated in selenoprotein synthesis in liver and particularly in SelP synthesis which acts as a major reservoir of Se and is mobilized from liver as necessary. It is therefore not expected that a regular input of Se from food alters Se levels in blood in the short term. That would only happen at a very high (and most likely toxic) dose of Se.

Figure 4 – Why would cooking affect partitioning of trace elements between blood compartments? once absorbed the behavior should be the same unless the absorption happened to be much different between raw and cooked and there could be saturation of ligands in one compartment in one of the scenarios. This was not the case so I don’t really see the need for such figure. A brief mention in the text would be enough.

Author Response

This is a relevant work providing much necessary in vivo data on the effect of cooking over MeHg bioavailability, since up-to-date most works were performed in vitro. In spite the low number of animals used authors compared animal data with in vitro experiments and clearly point the differences in bioavailability between the two different set-ups.

I have some comments/questions:

- From table 1 it seems cooked meals had higher concentration of MeHg, which is normal due to water loss. Why didn’t the authors adjust the mass given to pigs for the cooked meal to guarantee that they were getting the same amount of MeHg than the pigs that were fed the raw meal? Since the meal mass was the same between raw and cooked, the pigs getting cooked got more MeHg than the pigs getting raw meal.

We did adjust meal mass to take into account water losses during cooking. As indicated in the manuscript, pigs fed raw tuna meals received 495 ± 2.3 g while pigs fed with cooked tuna meals received 375 ± 0.4 g, in order to have similar MeHg doses (please see L149) after water loss. Differences of final doses between treatment reported in Table 1 likely result from the heterogeneity of MeHg distribution within tuna muscles.

Overall, there was on average 14% more MeHg in the cooked meals compared to the raw meals (see Table 1). A note about this 14% difference has been added (L150-152).

- Why wasn’t total Hg in serum analyzed by the same method as Hg in total blood? Was it related with the sample amount and the need to analyze also Se and As? Please clarify.

Clarifications have been made at L215-217.

- Lines 472-475: There seems to be a confusion here. If cooking decreases viscosity and therefore gastric empty is faster, the two variables are negatively correlated (as on decreases the other increases) and not positively as written

The authors agree. Modification has been made. (L511)

Lines 491 – 494: Keep in mind the absorbed Se (regardless it is inorganic or organic) does not go free into circulation. It is largely integrated in selenoprotein synthesis in liver and particularly in SelP synthesis which acts as a major reservoir of Se and is mobilized from liver as necessary. It is therefore not expected that a regular input of Se from food alters Se levels in blood in the short term. That would only happen at a very high (and most likely toxic) dose of Se.

The authors thank the reviewer for the information and have added this information in the manuscript (L538-541)

Figure 4 – Why would cooking affect partitioning of trace elements between blood compartments? once absorbed the behavior should be the same unless the absorption happened to be much different between raw and cooked and there could be saturation of ligands in one compartment in one of the scenarios. This was not the case so I don’t really see the need for such figure. A brief mention in the text would be enough.

Because of the lack of published information on the impact of cooking on metal partitioning in blood, we elect to keep the figure, and have added a clearer justification for the relevance of these results on L551-554